# The Levels of Cortisol and Selected Biochemical Parameters in Red Deer Harvested during Stalking Hunts

**DOI:** 10.3390/ani14071108

**Published:** 2024-04-04

**Authors:** Katarzyna Dziki-Michalska, Katarzyna Tajchman, Sylwester Kowalik, Maciej Wójcik

**Affiliations:** 1Department of Animal Ethology and Wildlife Management, Faculty of Animal Sciences and Bioeconomy, University of Life Sciences in Lublin, Akademicka 13, 20-950 Lublin, Poland; katarzyna.michalska@up.lublin.pl; 2Department of Animal Physiology, Faculty of Veterinary Medicine, University of Life Sciences in Lublin, Akademicka 12, 20-033 Lublin, Poland; 3Regional Directorate of the State Forests in Lublin, Czechowska 4, 20-950 Lublin, Poland; maciej.wojcik@lublin.lasy.gov.pl

**Keywords:** *Cervus elaphus*, stress response, stress hormone, biochemical parameters

## Abstract

**Simple Summary:**

The period of intense stalking hunting is recognized as a strong stress factor for game animals. Therefore, the level of stress hormone—cortisol, as well as other selected biochemical parameters—was assessed in the blood plasma of male and female red deer (*Cervus elaphus*) harvested during stalking hunts. It has been shown that this hunting method has a similar impact on cortisol levels in both bulls and hinds, and age did not affect its magnitude. However, there was an increase in some biochemical parameters (HDL cholesterol, lactate dehydrogenase, and alanine aminotransferase) in the blood of hinds and (alkaline phosphatase, bilirubin, aspartate aminotransferase) in the blood of bulls. It was also found that an increase in the concentration of stress hormone contributes to a reduction in carcass weight, which may suggest that increased stress negatively affects the body condition of game animals. To sum up, stalking hunts cause stress in red deer, which is reflected in changes in the plasmatic level of cortisol and some biochemical blood parameters. An additional important aspect of this research is to draw attention to the need to improve hunting methods in order to minimize stress in game animals as much as possible.

**Abstract:**

As a reactive species, the red deer is sensitive to both negative exogenous and endogenous stimuli. An intensive hunting period may have a particularly negative impact on game animals. The aim of this study was to determine the plasma cortisol level and biochemical parameters in 25 wild red deer (*Cervus elaphus*) harvested during stalking hunts in correlation with the sex and age of the animals. The mean cortisol concentrations in the stags and hinds analyzed in this study were similar (20.2 and 21.5 ng/mL, respectively). Higher HDL cholesterol values were found in the blood of the hinds than in stags (*p* < 0.05). Similarly, the mean levels of LDL cholesterol, lactate dehydrogenase, and alanine aminotransferase were higher by 21%, 16%, and 42%, respectively, in the blood of the hinds. In contrast, the levels of alkaline phosphatase, bilirubin, and aspartate aminotransferase were higher in the stags (by 30%, 49%, and 36%, respectively). There was a negative correlation of the cortisol concentration with urea and bilirubin and a positive correlation between cortisol and aspartate aminotransferase in the stags (*p* < 0.05). In turn, a negative correlation was found between the cortisol and urea levels in the hinds (*p* < 0.05). In summary, the stress caused by stalking hunts and the characteristic behavior of red deer during the mating season had an impact on chosen biochemical parameters. The increased concentration of cortisol resulted in a decrease in the carcass mass, which may lead to the deterioration of the physical condition of animals on hunting grounds.

## 1. Introduction

Deer hunting plays a multifaceted role in the USA and EU. Wildlife managers depend on hunting to manage the size of deer herds. Due to many causes, the deer numbers in some regions of the EU and USA have been increasing for years. Deer hunting also makes an important contribution to the economies of rural communities in the USA and Europe. The U.S. Fish and Wildlife Service has estimated that hunting and fishing generate more than 3 billion dollars in economic activity annually in New York (1996), but the EU estimated that hunting is worth about €16 billion annually (2008) [1,2].

There are four species of deer in Poland, and each of the following is a game species. The largest representative is the moose (*Alces alces*), which numbers 37,493 individuals. In turn, the most numerous cervid in Poland is the European roe deer (*Capreolus capreolus*), which numbers 893,200 individuals. The least numerous species observed and monitored in Poland is the fallow deer (*Dama dama*). The current population of red deer (*Cervus elaphus*) in Poland comprises 292,000 individuals [3]. Over the past five decades, the European population size and harvesting of this species have been increasing [3]. The red deer is represented by many subspecies inhabiting various ecosystems [4]. All cervids are sensitive to strong negative stimuli from the external environment [3,4,5], which exert a negative effect on the health and condition of these animals [6,7,8,9]. The impact of negative factors on the animal’s organism can be analyzed via the determination of the concentration of cortisol (CORT), a hormone associated with general stress reaction [10]. The welfare of deer is influenced by many exogenous factors, including climate, season, population density in a given area (which translates into access to food), or the presence of large predators hunting deer in the natural environment. Previous research has clearly shown that estrus is a particularly stressful period for deer due to the need for males to defend their territory and constant competition for females [11,12,13]. Negative stimuli also affect the growth and mineralization of antlers [14,15,16]. Nevertheless, animals subjected to selective culling are usually in good physical condition, even though they are constantly exposed to negative stimuli from the external environment. Gentsch et al. [11] indicate that although negative factors may accumulate and promote increased CORT release, the stimuli that determine the strength of the hormonal reaction during hunting are the short period before shooting (e.g., without long-time tracking) and the time from the hit to the death of the animal.

The concentration of CORT in hair, feces, urine, saliva, or plasma is frequently determined to assess the level of stress in animals [17]. Determining plasmatic CORT levels is widely used in monitoring acute stress reactions, whereas excretions (feces, urine) are more useful in the case of chronic, long-term stress reactions. The undoubted advantage of measuring the level of glucocorticoids in feces is the ability to collect a sample without the need to capture animals, which in turn eliminates unnecessary stress associated with immobilizing them or vein-puncturing to draw blood samples. Therefore, determining the concentration of cortisol metabolites in feces seems to be the most valuable method for studies on wild animals, where such procedures in the field may be difficult to perform [18,19]. Despite its advantages, this method also has some limitations, as it is often not possible to attribute feces to particular individuals, especially those living in large areas, without close monitoring. Therefore, if it is not possible to examine select individuals, the results obtained should be treated as collective for the population of animals inhabiting a given area. In red deer, maximum levels of CORT metabolites are usually measured after 18 h of stressor exposure based on fecal samples [12]. CORT can also be determined from animal hair samples as an indicator of stress conditions. However, it should be remembered that the level of glucocorticoid metabolites in hairs increases quite slowly, i.e., several weeks or months, depending on the species studied [20,21,22,23,24,25]. Hence, due to the limitations described above, in hunted animals, the best method seems to be rapid post-mortem blood collection, in which the CORT level shows the severity of acute stress [7,26,27,28,29].

Studies of wild animals are particularly difficult due to the fact that there are many factors that may affect the level of glucocorticoids obtained in the samples tested, such as ambient temperature (climate), food availability, and season [3,30,31]. These conditions, in turn, significantly affect the body’s energy balance, a key determinant influencing the animal’s ability to effectively respond to various stressors [32]. On the other hand, accumulated energy resources improve the physical condition and health status of animals, which can also influence the level of CORT [33,34].

CORT represents a group of steroid hormones and is released from the adrenal cortex in response to a negative stimulus through the activation of the hypothalamic–pituitary–adrenal (HPA) axis [35]. Changes in the circulatory concentration of CORT in red deer may have far-reaching effects [4,36]. One of the negative effects of stress may include deterioration of immunity [6,37], worsening of general condition, and enhancement of the susceptibility of the animal to various diseases [38]. An increase in the level of glucocorticosteroid hormones also disturbs the antler-shedding cycle in red deer [39,40] and is associated with proper antler growth [15,16]. Additionally, it may lead to myocardial myopathy and exercise-induced rhabdomyolysis, which is often documented in wild animals [41]. Other conditions that may affect CORT release may be related to intense short-term stress from transportation, immobilization, or predator attacks [42,43,44,45,46]. Stress also exerts an adverse effect on an animal’s body weight, which is one of the most important determinants of reproductive success [47]. Stress in hinds may have negative effects on the normal regulation of the estrus cycle and maintenance of pregnancy [48]. As demonstrated in studies on Canadian red deer (*Cervus elaphus canadensis*), CORT is positively linked with the level of progesterone secreted from adrenal glands in response to short-term stress [49].

Game animals are dependent on the natural environment because they are inextricably linked to it, and any change in their natural environment immediately affects the populations they create. Energy resources in the form of adipose tissue, antler mass, population density, and structure, but also biochemical and physiological parameters of the body, reflect the body’s internal balance and thus allow for maintaining good well-being, which is influenced by various factors of the external environment [50,51,52,53]. Blood biochemical parameters act as markers of potential disease because their fluctuations outside the reference limits indicate disturbances in the body’s homeostasis, which may be related to the ongoing disease process [4,48] and the impact on the behavior and mental state of the animal, i.e., [6,52]. It is generally known that hunting is a strong stressor in wild animals, which is associated with short-term (fight-or-flight) stress responses [54]. However, due to the seasonality of hunting and the related intensification of human activity in the cervid ecosystem, wild animals may be affected by chronic stress [55,56]. Some authors find that the stress response of the animal organism depends on the type of hunting [11,55]. However, it is worth noting that Ensminger et al. [56] found no evidence of increased fecal glucocorticoids in association with the number of days of hunting, even though fecal glucocorticoids concentration increased with hunting pressure index, as previously shown in *Cervus elaphus* [57]. On the other hand, Bateson and Bradshaw’s [57] study was conducted during long red deer pursuits (about 19 km) using hunting dogs, and the increased glucocorticoid levels in the collected blood samples likely reflected the last pursuit [58]. In our research, hunters may have harvested a *Cervus elaphus* soon after observing it on the first day without the use of dogs or actively driving it into the hunting ground [57,58]. These studies make it possible to compare the impact of indirect and long-term hunting with direct, short-term hunting on the CORT level in the feces and blood serum of game animals. It made it possible to compare the indirect and prolonged impacts of hunting pressure on red deer, as indicated by fecal glucocorticoids [59,60,61].

In Poland, red deer, in addition to collective hunting, are quite often obtained by stalking. The impact of these hunts can be considered at two levels: short-term stress caused at the time of the shooting and long-term stress caused by the frequent presence of hunters in the hunting ground. The impact of stalking has rarely been studied until now. Therefore, the aim of this study was to determine the plasma cortisol level in wild red deer (*Cervus elaphus*) harvested during stalking hunts in relation to the sex and age of the animals. In addition, the health status of the red deer was determined on the basis of selected blood biochemical parameters, which may affect the level of circulatory cortisol and vice versa.

## 2. Materials and Methods

### 2.1. Experimental Design

The tested red deer were hunted in accordance with Polish Hunting Law and the principles of individual selection during the hunting season as part of population management. This law states that red deer stags can be hunted from 1 August to the end of February and hinds from 1 September to 15 January. The individuals covered by the research were obtained in the period from 1 September to 31 October 2022, and the shooting of animals had a selective purpose and was carried out by qualified hunters—employees of the National Lubartów Forests District. In order to obtain animals, individual hunts were carried out, which involved approaching the deer unnoticed at a distance, allowing for a precise shot without the battue or hunting dogs. The hunting took place in the early morning hours, during which the animals behaved naturally, showed no signs of stress, and were not artificially separated from the herd. The animals selected for this study were individuals that died immediately after the first precise shot.

The Lubartów Forest District is located in central–eastern Poland (51°27′ N, 22°29′ E). The terrain of the district is flat rather than varied and has a temperate to warm and transitional climate. Annual rainfall is 552 mm, and the average annual temperature is +7.7 °C. The forest district is characterized by a large diversity of habitats and soil fertility. The forest cover of the region is estimated at 24.9%, including 49% of the area covered by coniferous species and 38% covered by mixed forests with deciduous species [62].

In the study area, the density of red deer was 16.03 individuals per 1000 ha of forest area, the density of roe deer was 2.08 individuals per 1000 ha of total area, and the density of moose was 10.15 individuals per 1000 ha of forest and swampy areas (as of 31 January 2017) [62].

As was previously mentioned, stalked animals were shot in accordance with the Polish Hunting Law (annex to Resolution No. 57/2005 of 22 February 2005). Due to the fact that all hunting procedures took place as part of hunting management in the areas managed by the State Forest District in Lubartów, no additional experimental procedures were performed that would pose a threat to animal welfare. Therefore, this study did not require individual consent from the Local Ethics Committee (Animal Welfare Committee Regulations, Faculty of Animal Sciences and Bioeconomy, University of Life Sciences in Lublin, ZdsDz/6/2023).

### 2.2. Sampling

Blood samples were collected from 13 stags and 12 hinds immediately after shooting and up to five minutes after death. Venous blood samples were drawn from the external jugular vein (vena jugularis externa) into 10 mL EDTA vacuum tubes (BD Vacutainer System, Ref. No. 367525, Becton Dickinson, Poland, Warsaw) and then cooled to 4 °C. Contaminated blood samples were discarded. Plasma was obtained by centrifugation of the blood samples at 3000 rpm for 10 min in a laboratory centrifuge, MPW-350R (MPW Medical Instruments, Warsaw, Poland), at 4 °C. Harvested plasma was stored at −25 °C until further analyses. CORT concentration was determined using the immunoenzymatic method (Cortisol ELISA Kit, No. EIA-1887, DRG^®^International, Springfield, NJ, USA) in accordance with the protocol recommended by the manufacturer. Biochemical parameters: total cholesterol (TCHOL), HDL cholesterol (HDLCHOL), LDL cholesterol (LDLCHOL), triglycerides (TRIG), lactate dehydrogenase (LDH), urea (UREA), alanine aminotransferase (ALAT), uric acid (URIC), phosphatase alkaline (ALP), total protein (TP), albumin (HSA), bilirubin (BIL), aspartate aminotransferase (ASAT), and gamma-glutamyl transpeptidase (GGTP) were determined using an automated spectrophotometric system (Biochemical analyzer BS-120, Mindray, Shenzhen, China). The age (A) of the animals was determined post-mortem using the Eidmann method. It consisted of an assessment of the layers of dentin deposited in the canal of the first pair of incisors I1 and the characteristic features of the dentition, i.e., the stage of development and the replacement of primary with permanent teeth [57]. Carcass mass (CM) was measured after the animals were harvested and eviscerated at a game collection center.

### 2.3. Statistical Analysis

The statistical analyses were conducted using the Statistica 9.1 software (StatSoft, Kraków, Poland). A significance level of *p* < 0.05 was adopted to indicate the existence of statistically significant differences or relationships. The normality of the distribution was tested using the Student’s *t*-test and the Welch test. The Mann–Whitney test was used for the analysis of independent samples. The results were expressed as the mean value (M) and standard deviation (SD) of the variables. Next, the CORT concentration was compared with the level of biochemical parameters in the sex-related groups and in the group of all animals using Spearman’s rank–order correlation. Subsequently, the correlations of the carcass mass of the harvested red deer with the level of CORT and biochemical parameters were calculated using an analogous method. Similar calculations were conducted to determine the relationship of the animal age with the concentration of CORT and the biochemical blood parameters. The analyses were carried out both in the group of all animals and in the sex-related groups.

## 3. Results

The mean CM of the stags (138.429 kg) was almost twice as high as that of the hinds (71.000 kg). The stags were aged from 5 to 8 years, while the hinds were aged from 3 to 5 years, with an average age of 6 and 4 years, respectively. The average CORT concentration in the plasma of the stags and hinds was similar (20.216 and 21.564 ng/mL, respectively). Significantly higher HDLCHOL values were determined in the plasma collected from the hinds than stags (25.678 and 9.114 mg/dL, respectively). Similarly, the mean levels of LDLCHOL, LDH, ALAT, and GGTP were 21%, 16%, 42%, and 39% higher, respectively, in the plasma of the hinds. In turn, the levels of ALP, BIL, and ASAT were higher (by 30%, 49%, and 36%, respectively) in the plasma collected from the stags (Table 1).

Next, the mean level of cortisol in correlation with the values of the biochemical parameters of the blood collected from the stags and hinds was analyzed statistically (Table 2). In the stag blood, there was a negative relationship between CORT with UREA and BIL (*p* < 0.05) and a positive correlation between CORT and ASAT (*p* < 0.05). In turn, a negative correlation was found between CORT and UREA in the hind blood (*p* < 0.05). Regardless of the sex of the animals, there was a negative correlation of CORT with TRIG, UREA, TP, and HSA (*p* < 0.05).

The next step consisted of the analysis of the relationship of the age (A) and carcass mass (CM) with the biochemical parameters (Table 3). In the group of all animals, only CM and HDLCHOL exhibited a negative correlation (*p* < 0.05) (Table 3).

No statistical correlations between CM, A, and biochemical parameters were found in the group of stags (Table 4).

A positive correlation between CM and HDLCHOL was found in the group of hinds (*p* < 0.05) (Table 5).

## 4. Discussion

All types of hunting affect the homeostasis of game animals and are perceived as a highly stressful stimulus [55]. Sheriff et al. [27] reported that increased physical activity in animals that were hunted evoked a typical stress response accompanied by an increase in the blood CORT concentration. However, this increase was not as high as in the case of a severe injury inflicted by a car collision, where the concentration of this hormone was significantly higher than that reported in previous studies [27]. It should be emphasized that the increase in CORT is also influenced by the fact that the animal feels mental discomfort, correlated with the feeling of physical pain, fear, or panic in response to various negative stimuli [55]. Moreover, research by Malherbe [63] showed that an increased level of CORT may even be the result of being too close in proximity to the anthropogenic environment (buildings and human settlements). The results obtained in the present study on red deer harvested during stalking hunts confirm previous reports showing that CORT levels do not depend on the age and sex of animals [5,12,64,65,66,67,68]. It could indicate that, regardless of age or sex, being hunted is a stressful experience for red deer. However, the absence of age-related differences in the CORT concentrations calls into question the reports indicating that older stages have higher CORT levels [23,69]. The mean CORT level shown in the present study was higher than that presented by Gaspar-López et al. [16] in farmed Iberian red deer (*Cervus elaphus hispanicus*) in the same season of the year. As reported in the literature, the concentration of plasma CORT is subject to both diurnal and annual fluctuations [70,71,72]. Ingram et al. [72] showed seasonal fluctuations in the CORT concentration in a fairly wide range from 1.9 to 22.5 ng/mL in unrestrained farmed red deer (*Cervus elaphus*), with higher levels recorded in November during the period of intensive weight gain after the mating season. These observations confirm earlier findings reported by Saltz and Whitey [4]. Our results are also consistent with those obtained by Ingram et al. [72], but they are more within their upper limits, which may be related to the mating season and not to body weight gain.

The mean levels of most biochemical parameters did not differ significantly between the groups of the stages and hinds. There was a significant difference in the average level of HDL CHOL, which was almost threefold higher in the hinds. In turn, the higher levels of LDLCHOL, LDH, ALAT, and GGTP in the hind blood plasma than those in the blood of stags did not significantly differ statistically. Slight differences were found in the ALP, BIL, and ASAT levels, which were higher in the blood samples from the stags. The high level of cholesterol fractions may be associated with the fact that these compounds are involved in the synthesis of steroid hormones, which are an important link in the regulation of physiological processes in the organism [73,74]. Studies conducted on male Iberian deer (*Cervus elaphus hispanicus*) in two different populations with high and low levels of intrasexual competition showed a positive relationship between the levels of testosterone and cortisol metabolites in feces but also a meaningful interaction, showing that this relationship occurs more intensely within a population characterized by high level of competition for mating. These results confirm a positive relationship between both hormones in natural conditions and that the reason for this is competition by male-to-male competition for mates [75].

It should be noted that the red deer were harvested during the mating season, which undoubtedly had an impact on the concentrations of the biochemical parameters determined in this study. The level of testosterone in the testicles of wild stags in September is 1000-fold higher than in the other months [76]. Physiologically, testosterone is synthesized from cholesterol; thus, these parameters have been reported in the literature to be significantly positively correlated [77,78]. In turn, high concentrations of individual cholesterol fractions in red deer plasma may cause serious health consequences [79]. Finally, the maintenance of the dominant male position in a herd of farmed hinds also generates a strong stress reaction [80], which may be reflected in the parameters studied.

Elevated CORT and LDH levels indicate potential muscle damage caused by various trauma factors [44,81]. Furthermore, the level of biochemical parameters may be influenced by cervid nutrition [82]. The red deer were harvested during the mating season—a period when they drastically reduce food intake [83]. Fasting, in turn, most likely leads to a decrease in some biochemical parameters (TCHOL, HDLCHOL, LDLCHOL, LDH, TP, GGTP) in stags but not in hinds, which do not exhibit such behavior [84]. Moreover, it should be noted, however, that the difference in time from shooting to death of the animals may affect the results of biochemical parameters of the blood serum. 

Generally, the welfare of free-living animals depends on exogenous and endogenous factors. The endogenous factors include the general health status of the animal and the impact of environmental stressors, e.g., the season of the year, food availability, reproduction, predators, and hunting [5]. The levels of blood biochemical parameters demonstrated in the present study were higher than those reported by other researchers in their studies on farmed red deer (*Cervus elaphus*) [6,85,86,87] and were also dependent on the type of physical or chemical anesthetic used and/or sedation agents used to subdue the animals [88] (Table 6). Our results do not align with the ranges indicated by Rosef et al. [89], who reported higher levels of AST, GGTP, LDH, UREA, TCHOL, and TRIG and lower levels of TP, HAS, and BIL in wild *Cervus elaphus atlanticus* pharmacologically immobilized. Nevertheless, it should be taken into account that these discrepancies may stem from the differences in environmental factors, e.g., the climate prevailing in the habitats of the analyzed populations, the methods (pharmacologically, physically immobilized, or shot) for collection of biological material for analyses [4,12,90], and the higher wild animal population density [67] (Table 6).

The obtained results showed that the CORT concentration in the plasma was negatively correlated with UREA and BIL in the group of stags and only with UREA in the group of hinds. It is worth noting that the level of UREA may be influenced indirectly by the season of the year and directly by the inaccessibility of food, as demonstrated in sheep [91] as well as in cervids [30,92]. Malnutrition leads to protein catabolism and tissue disintegration, which is reflected in an increase in UREA levels in animal blood [93]. In the present study, much higher values in the UREA of all the analyzed red deer were obtained compared to the results obtained by Rosef et al. [89] for free-ranging animals, which may be related to the insufficient protein intake due to the reduction of appetite during the mating season [94]. Physiologically, an increase in UREA is observed during stress, increased catabolism, and post-traumatic conditions [95]. However, in our studies, lower UREA values were observed, which may indicate that the response of this parameter to hunting is a slower reaction compared to the rapid increase in CORT. Another reason for low UREA values may be the seasonal change of animals’ diet to winter when vegetation low in protein and rich in carbohydrates dominates, and as it is generally known, the level of feed protein affects the level of plasmatic UREA. It has also been documented that during pregnancy, the renal glomerular filtration rate increases, which causes increased secretion of UREA in relation to its production [96]. This study did not assess the reproductive status of females; however, considering the fertile period of cervids, it can be assumed that over 80% of females could have been pregnant [55]. Nutritional stress may also have an impact on the level of BIL, as shown in studies on *Lepus californicus* [97]. In addition, the obtained results showed a strong negative correlation between the parameters in the blood of the stags, e.g., an increase in the CORT concentration was accompanied by a decrease in the BIL level. A positive correlation between CORT and ASAT was found in the group of stags as well. ASAT is a muscle tissue-specific enzyme. Its increased activity is directly related to muscle damage [98,99]. ASAT also serves as an indicator of liver disease, as its levels increase before liver degeneration, which becomes apparent in clinical trials [100], and the increase may be caused by the cessation of feed intake by stags during the rutting season.

In studies on red deer, low TP and TRIG confirm the reduced food intake by the animals during the mating season, as both of these parameters are considered markers of malnutrition [101,102,103]. Also, the negative correlation between CORT and URIC in the groups of stags, hinds, and all animals taken together is not surprising, as URIC is considered a marker of long-term stress [104]. Moreover, these results were directly linked to the rutting season, i.e., the sampling period, which was a long-term negative stimulus for the animals. A negative correlation between BIL and A was found in the group of the studied hinds. Nevertheless, the BIL concentration was similar to that determined by Alhuay et al. [105] in white-tailed deer (*Odocoileus virginianus*) but lower than that specified by the ISIS standard [106]. However, the concentration of BIL depends on many factors, e.g., the latitude, climate, and determination methods [105].

Also, the analyses of A and CM were performed with reference to the biochemical parameters. A positive correlation between CM and HDLCHOL was demonstrated in the group of hinds. This is a disturbing phenomenon, as the examined hinds were relatively young, and female steroid hormones are known to be synthesized from CHOL, whose proper level has an impact on the proper reproduction processes of this species [48,107]. Additionally, the negative correlation of CORT with CM indicates that the higher the cortisol concentration in red deer blood, the lower the carcass mass, which may directly influence the condition of these animals [52]. However, this assumption should still be confirmed by methods relating to long-term stress, e.g., measuring the level of glucocorticoid metabolites in animal hair [108].

## 5. Conclusions

Plasma CORT levels in red deer harvested during stalking hunts were similar in males and females and amounted to an average of 20.216 ng/mL and 21.564 ng/mL, respectively. The variability of the analyzed blood biochemical parameters was probably associated with the increased concentration of the CORT and, thus, the sensitivity of the species regardless of age or sex. The mating season, which is known for increased activity, changed the diet and physiological state of animals and had an additional effect on some biochemical parameters (TCHOL, HDLCHOL, LDLCHOL, LDH, TP, GGTP). Moreover, with the increase in CORT and UREA and decrease in BIL, ASAT increased in red deer in the study period. The higher concentration of cortisol exerted a negative impact on carcass mass, which may determine the deterioration of the general condition of the red deer.

Repeated exposure of red deer populations to game hunting has an impact on stress levels, which may have important implications for the sustainability and conservation of this species. In particular, stress can influence population dynamics by altering feeding and breeding behavior, animal welfare, and, ultimately, evolutionary processes, altering individual adaptation and selection. 

## Figures and Tables

**Table 1 animals-14-01108-t001:** Dependence of the carcass mass (CM), age (A), and biochemical parameters in plasma on the sex of the red deer.

Indicator	Short	Male (n = 13)		Female (n = 12)		T ^a^/U ^b^	*p*
		M	SD	M	SD		
Carcass mass [kg]	CM	138.42	20.75	71.00	11.84	8.22843 ^a^	<0.001 *
Age [years]	A	6	2.050	4	1.000	2.48455 ^a^	0.026 *
Cortisol [ng/mL]	CORT	20.216	16.747	21.564	17.325	29.000 ^b^	0.837
Total cholesterol [mg/dL]	TCHOL	85.714	65.642	108.889	53.412	16.000 ^b^	0.114
Cholesterol HDL [mg/dL]	HDLCHOL	9.114	5.712	25.678	8.541	−4.4051 ^a^	<0.001 *
Cholesterol LDL [mg/dL]	LDLCHOL	20.714	20.798	25.667	10.062	17.500 ^b^	0.141
Triglycerides [mg/dL]	TRIG	290.429	315.265	251.333	155.251	23.000 ^b^	0.407
Lactate dehydrogenase [U/L]	LDH	3609.714	2204.350	4160.444	3461.297	27.000 ^b^	0.680
Urea [mg/dL]	UREA	53.629	24.231	51.889	12.212	0.188 ^a^	0.853
Alanine aminotransferase [U/L]	ALAT	235.714	174.545	333.378	163.478	−1.151 ^a^	0.268
Uric acid [mg/dL]	URIC	21.243	23.550	18.700	17.765	26.000 ^b^	0.606
Alkaline phosphatase. [U/L]	ALP	157.029	147.178	111.267	84.550	0.785 ^a^	0.445
Total protein [g/dL]	TP	10.571	4.826	12.000	3.969	19.500 ^b^	0.210
Albumin [g/dL]	HSA	4.286	1.254	4.333	0.707	27.500 ^b^	0.680
Bilirubin [mg/dL]	BIL	0.901	0.837	0.454	0.338	1.331 ^c^	0.221
Animotransferaza asparaginianowa [U/L]	ASAT	166.529	153.196	106.089	68.339	1.063 ^a^	0.305
Gamma-glutamylotranspeptydaza [U/L]	GGTP	61.357	20.735	99.667	92.720	28.000 ^b^	0.757

M—mean; SD—standard deviation; ^a^—the Student’s *t*-test result; ^b^—Mann–Whitney test results; ^c^—Welch’s *t*-test results; * statistically significant values at *p* < 0.05.

**Table 2 animals-14-01108-t002:** Relationship between the cortisol concentration and the biochemical parameters of the blood of the analyzed animals.

Analyzed Parameters	CORT		
	Male (n = 13)	Female (n = 12)	All Animals (n = 25)
	r-Spearman’s rank–order correlations; *p*-Value		
TCHOL [mg/dL]	−0.450; 0.310	−0.450; 0.310	−0.441; 0.086
HDLCHOL [mg/dL]	−0.036; 0.938	−0.036; 0.938	0.041; 0.879
LDLCHOL [mg/dL]	−0.560; 0.190	−0.560; 0.190	−0.384; 0.141
TRIG [mg/dL]	−0.468; 0.288	−0.468; 0.288	−0.529; 0.034 *
LDH [U/L]	0.684; 0.089	0.684; 0.089	0.281; 0.291
UREA [mg/dL]	−0.846; 0.016 *	−0.846; 0.016 *	−0.563; 0.022 *
ALAT [U/L]	0.234; 0.613	0.234; 0.613	0.022; 0.935
URIC [mg/dL]	−0.594; 0.159	−0.594; 0.159	−0.459; 0.073
ALP [U/L]	−0.234; 0.086	−0.266; 0.487	−0.420; 0.104
TP [g/dL]	−0.441; 0.613	−0.630; 0.068	−0.668; 0.004 *
HSA [g/dL]	−0.715; 0.089	−0.547; 0.126	−0.615; 0.011 *
BIL [mg/dL]	−0.918; 0.003 *	0.133; 0.732	−0.280; 0.292
ASAT [U/L]	0.756; 0.048 *	−0.233; 0.545	0.266; 0.318
GGTP [U/L]	−0.702; 0.078	−0.183; 0.636	−0.400; 0.124

* statistically significant values at *p* < 0.05.

**Table 3 animals-14-01108-t003:** Relationship between the level of biochemical parameters, carcass mass, and age in the group of all animals.

Analyzed Parameters	Carcass Mass (CM)	Age (A)
	r/R; *p*-Value	
CORT [ng/mL]	−0.116 ^a^; 0.666	−0.108 ^a^; 0.689
TCHOL [mg/dL]	−0.339 ^a^; 0.198	−0.057 ^a^; 0.832
HDLCHOL [mg/dL]	−0.663 ^a^; 0.005 *	−0.442 ^b^; 0.086
LDLCHOL [mg/dL]	−0.354 ^a^; 0.177	−0.009 ^a^; 0.973
TRIG [mg/dL]	−0.131 ^a^; 0.627	0.194 ^a^; 0.470
LDH [U/L]	0.013 ^a^; 0.961	−0.044 ^a^; 0.868
UREA [mg/dL]	−0.135 ^a^; 0.615	0.103 ^b^; 0.702
ALAT [U/L]	−0.316 ^a^; 0.232	0.094 ^b^;0.727
URIC [mg/dL]	−0.221 ^a^; 0.409	0.216 ^a^;0.419
ALP [U/L]	−0.056 ^a^; 0.836	0.266 ^a^; 0.318
TP [g/dL]	−0.281 ^a^; 0.290	−0.003 ^a^; 0.991
HSA [g/dL]	−0.012 ^a^; 0.962	0.176 ^a^; 0.513
BIL [mg/dL]	0.207 ^a^; 0.440	0.117 ^a^; 0.665
ASAT [U/L]	0.234 ^a^; 0.381	−0.008 ^a^; 0.973
GGTP [U/L]	0.067 ^a^; 0.802	0.210 ^a^; 0.432

^a—^Spearman’s rank–order correlations; ^b—^Pearson r correlations; * statistically significant values at *p* < 0.05.

**Table 4 animals-14-01108-t004:** Relationship between the level of biochemical parameters, carcass mass, and age in the group of stags (group I).

Analyzed Parameters	Carcass Mass (CM)	Age (A)
	r-Spearman’s rank–order correlations; *p*-Value	
CORT [ng/mL]	−0.018; 0.969	0.284; 0.536
TCHOL [mg/dL]	0.126; 0.787	0.090; 0.846
HDLCHOL [mg/dL]	−0.468; 0.288	−0.054; 0.907
LDLCHOL [mg/dL]	0.000; 1.000	0.094; 0.840
TRIG [mg/dL]	0.126; 0.787	0.290; 0.526
LDH [U/L]	−0.090; 0.847	0.436; 0.327
UREA [mg/dL]	0.018; 0.969	0.218; 0.638
ALAT [U/L]	−0.072; 0.877	0.436; 0.327
URIC [mg/dL]	0.036; 0.938	0.290; 0.526
ALP [U/L]	−0.720; 0.067	0.272; 0.553
TP [g/dL]	0.336; 0.460	0.169; 0.715
HSA [g/dL]	0.381; 0.398	−0.0481; 0.918
BIL [mg/dL]	0.336; 0.460	−0.110; 0.814
ASAT [U/L]	0.018; 0.969	−0.327; 0.473
GGTP [U/L]	−0.072; 0.877	−0.600; 0.154

**Table 5 animals-14-01108-t005:** Relationship between the level of biochemical parameters, carcass mass, and age in the group of hinds (group II).

Analyzed Parameters	Carcass Mass (CM)	Age (A)
	r-Spearman’s rank–order correlations; *p*-Value	
CORT [ng/mL]	−0.238; 0.536	−0.053; 0.891
TCHOL [mg/dL]	0.183; 0.635	0.454; 0.219
HDLCHOL [mg/dL]	0.689; 0.039 *	−0.195; 0.615
LDLCHOL [mg/dL]	0.004; 0.991	0.645; 0.060
TRIG [mg/dL]	0.195; 0.613	0.531; 0.140
LDH [U/L]	−0.349; 0.357	−0.354; 0.348
UREA [mg/dL]	−0.170; 0.661	−0.062; 0.873
ALAT [U/L]	0.008; 0.982	0.461; 0.211
URIC [mg/dL]	−0.417; 0.264	0.398; 0.287
ALP [U/L]	0.008; 0.982	0.328; 0.388
TP [g/dL]	−0.012; 0.973	0.049; 0.900
HSA [g/dL]	0.081; 0.834	0.315; 0.407
BIL [mg/dL]	−0.212; 0.582	−0.611; 0.079
ASAT [U/L]	0.042; 0.913	0.328; 0.388
GGTP [U/L]	−0.017; 0.965	0.691; 0.390

* statistically significant values at *p* < 0.05.

**Table 6 animals-14-01108-t006:** Biochemical parameters value in the earlier studies.

Analyzed Parameters	Wild Red Deer [90]		Chemically Immobilized Free-Ranging Red Deer at Winter Feeding Sites [89]	Farmed Red Deer		Wild Red Deer Harvested during Stalking Hunts (Our Study)	
	Physical Capture	Chemical Capture		[6]	[87]	Male	Female
TCHOL [mg/dL]	55.44 ± 14.93	50.25 ± 19.63	41.76–50.65	57.61 ± 13.14	59.9 ± 3.18	85.714	108.889
HDLCHOL [mg/dL]	34.44 ± 9.27	31.75 ± 10.30	-	-	41.58 ± 2.80	9.114	25.678
LDLCHOL [mg/dL]	3.75 ± 1.70	2.0 ± 1.0	-	-	15.00 ± 3.13	20.714	25.667
TRIG [mg/dL]	18.50 ± 7.41	7.88 ± 6.77	8.75–10.50	19.25 ± 9.62	16.90 ± 4.15	290.429	251.333
LDH [U/L]	511.75 ± 93.89	404.33 ± 120.35	672.5–741.2	-	-	3609.714	4160.444
UREA [mg/dL]	-	-	5.76–6.78	66.66 ± 19.09	-	53.629	51.889
ALAT [U/L]	-	-	51.7–56.7	33.5 ± 8.66	-	235.714	333.378
URIC [mg/dL]	0.32 ± 0.18	0.21 ± 0.03	-	-	0.37 ± 0.13	21.243	18.700
ALP [U/L]	144.550 ± 25.26	104.33 ± 20.24	195.5–252.1	31.6 ± 10.71	-	157.029	111.267
TP [g/dL]	6.77 ± 0.79	6.77 ± 0.99	6.36–6.63	8.08 ± 10.70	6.26 ± 0.36	10.571	12.000
HSA [g/dL]	-	-	3.57–3.75	2.15 ± 2.93	-	4.286	4.333
BIL [mg/dL]	0.12 ± 0.05	0.06 ± 0.04	47.95–64.96	599.38 ± 154.29	-	0.901	0.454
ASAT [U/L]	-	-	55.0–63.3	250.5 ± 117.44	-	166.529	106.089
GGTP [U/L]	20.77 ± 5.86	19.25 ± 4.57	18.4–22.4	15.5 ± 9.82	-	61.357	99.667

## Data Availability

The data that support the findings of this study are available on request from the corresponding author.

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
