# Peer review of "The Levels of Cortisol and Selected Biochemical Parameters in Red Deer Harvested during Stalking Hunts"

_animals, 2024, doi:10.3390/ani14071108_

Round 1

Reviewer 1 Report

Comments and Suggestions for Authors

The levels of cortisol and selected biochemical parameters in red deer harvested during stalking hunts were detected in this paper, and the results were analyzed in detail according to age and sex. Overall, this paper is generally well-written and conclusions will be helpful to the protection and hunting management of red deer. However, there are some issues to be addressed.

Lines 36-37: It is recommended to give a brief description of the strong negative stimuli to the cervids.

Line 148: Please explain the reason for using carcass mass instead of body weight ? Is there a corresponding standard for carcass mass determination?

Lines 263-276: Considering that there are many studies on the blood biochemical parameters of red deer, if possible, please add a table to compare the ranges or results obtained from different studies.

Lines 332-334: This study did not determine the blood biochemical parameters of red deer during the non-mating season, and this conclusion needs to be more cautious.

Lines 107, 185, 216, 230, 328: There are some formatting problems in these lines, such as incorrect use of space, punctuation, number of significant figures (Table 1) and decimal point.

Author Response

Reviewer 1

The levels of cortisol and selected biochemical parameters in red deer harvested during stalking hunts were detected in this paper, and the results were analyzed in detail according to age and sex. Overall, this paper is generally well-written and conclusions will be helpful to the protection and hunting management of red deer. However, there are some issues to be addressed.

Response: Thank you for your kind comments.

Lines 36-37: It is recommended to give a brief description of the strong negative stimuli to the cervids.

Response: It was added, please see lines 59-71.

Line 148: Please explain the reason for using carcass mass instead of body weight ? Is there a corresponding standard for carcass mass determination?

Response: When hunting, it is not possible to measure the body mass of shot animals, we can only measure mass the carcass, that is, after removing the viscera, which should be done as soon as possible to prevent scalding of the carcass. The carcasses prepared in this way were weighed in the cold store, using an available scale.

Lines 263-276: Considering that there are many studies on the blood biochemical parameters of red deer, if possible, please add a table to compare the ranges or results obtained from different studies.

Response: It was added, please see Table 6.

Lines 332-334: This study did not determine the blood biochemical parameters of red deer during the non-mating season, and this conclusion needs to be more cautious.

Response: It was deleted.

Lines 107, 185, 216, 230, 328: There are some formatting problems in these lines, such as incorrect use of space, punctuation, number of significant figures (Table 1) and decimal point.

Response: It was improved.

Reviewer 2 Report

Comments and Suggestions for Authors

The authors collected a series of physiological parameters on 25 red deers harvested during stalking hunts. While I recognise some value in this investigation, and the numerous data collected are interesting, I believe the authors did not really answer their question to assess the impact of staking hunts on red deers. They just collected data from hunted deers without knowing for how long they where stalked before killing and without a control population. So on that basis the authors need to rephrase most of the paper and draw different conclusions. Potentially they can look for expected parameters in wild populations without stalking hunting pressure (if any data is available). But at the moment all the narrative of staking hunting is not supported as 1) it is missing details on hunting tactics before killing; 2) it is missing a control population. at most what you have is sex differences in parameters and correlations with weight and age (and a bit weak since you did not get many significant results, sample size is too small). If you really want to use these data and narrative, you can look for published parameters on wild populations without hunting impact, still not optimal but at least with some support. It would be also better to run GLMs with al factors in one model as now the series of tables are too long. And also if you run multiple tests on the same variables you need to consider corrections of p-values.   

There are other issues, such as the introduction starting too specific on the context while there should be a broader first paragraph, but these would be less important issues to consider after the above issues are addressed. 

Comments on the Quality of English Language

na

Author Response

Reviewer 2

The authors collected a series of physiological parameters on 25 red deers harvested during stalking hunts. While I recognise some value in this investigation, and the numerous data collected are interesting, I believe the authors did not really answer their question to assess the impact of staking hunts on red deers. They just collected data from hunted deers without knowing for how long they where stalked before killing and without a control population. So on that basis the authors need to rephrase most of the paper and draw different conclusions. Potentially they can look for expected parameters in wild populations without stalking hunting pressure (if any data is available). But at the moment all the narrative of staking hunting is not supported as 1) it is missing details on hunting tactics before killing; 2) it is missing a control population. at most what you have is sex differences in parameters and correlations with weight and age (and a bit weak since you did not get many significant results, sample size is too small).

Response: In studies conducted on large game animals living in the wild, it is very difficult to create a classic control group. Animals can only be caught during the hunting season, in accordance with the Hunting Law. As a control group, it would be necessary to catch the animals in a net and use pharmacological sedation to collect blood from them for testing. This type of procedure would significantly increase the level of stress hormone, which would affect the obtained results. We have no other option to physically immobilize or collect blood for testing from wild deer. Therefore, no control group was created.

Statistically, a large research group is considered to be a group of 25 deer (including 13 stags and 12 hinds).

If you really want to use these data and narrative, you can look for published parameters on wild populations without hunting impact, still not optimal but at least with some support. It would be also better to run GLMs with al factors in one model as now the series of tables are too long. And also if you run multiple tests on the same variables you need to consider corrections of p-values. 

Response: Most of the analyzed features are continuous, therefore analyzes of relationships between continuous features - correlations - were used. The use of GLM would make more sense when analyzing fixed factors with a small number of levels. A solution may also be the use of mixed GLM models that take into account constant factors, e.g. gender and regression factors, including: cortisol level or age - the introduction of this model will, however, have no impact on the presented results and final conclusions. Therefore, we decided not to recalculate the statistics using GLM models.

There are other issues, such as the introduction starting too specific on the context while there should be a broader first paragraph, but these would be less important issues to consider after the above issues are addressed. 

Response: Thank you for your comment, however one of the reviewers even recommended expanding the "Introduction" chapter.

Reviewer 3 Report

Comments and Suggestions for Authors

In many places in the world, different species of deer are hunted for various reasons. As such this paper will be of interest to a wide audience. In addition, this is an important topic to confront because deer hunting exposes animals to pain, fear, stress, distress and, by definition, premature death. As such, this study makes an important contribution to the literature because it demonstrates the stress caused by stalking hunts. However, the manuscript raises several serious animal welfare concerns including failure to kill animals outright and the killing of pregnant females and unborn fawns, which would be contrary to legislation in other jurisdictions. It is my strong view that while the manuscript has merit, these and several other aspects of the paper require major revision to be considered for publication.

Introduction
Line 33 This is interesting background information particularly to readers who are unfamiliar with deer hunting in Poland and Europe more broadly. Suggest you expand to include several paragraphs:
- General intro on numbers & species (as you have provided lines 33-35)
- Hunting pressures, trends, methods, controversies in Poland & elsewhere
- The suite of potential stressors associated with hunting (line 109 you mention short-term stress at the time of shooting and long-term stress caused by presence of hunters but there are so many more potential stressors to both the hunted and the survivors (e.g., wounding, social disruption interrupted feeding and drinking etc.). If your central proposition is that [CORT] in blood collected immediately post-kill is an accurate reflection of the severity of acute stress, it’s highly relevant for readers to understand all the potential stressors that could be contributing to the CORT spike you predict.
- The sensitivity of deer to negative stimuli including those associated with hunting: this is a key point that deserves a paragraph in your introduction. Draw on literature about behavioural reactivity of cervids e.g., freeze, flight, their propensity to panic. Brief statement of the relevant behavioural ecology would be valuable too i.e., why deer are like this. Anyone who has worked with captive or wild deer has seen them react so dramatically to stimuli that they’ve caused themselves fatal injuries like running into walls, fences etc.  
- Stress physiology (as you have provided lines 39 onwards)

Line 49 Typo ‘in the field’ instead of ‘in the filed’

Line 61 Would be pertinent to add ‘exercise’ or ‘physical exertion’ to this list as you go on to discuss this later in the paper  

Line 87 It is meaningful here that you have highlighted CORT as an animal welfare indicator but this section would be strengthened if you not only talk about physical state e.g., (“potential disease states”) but also the impact on the animal’s behavioural and mental state i.e., the Five Domains of Animal Welfare 
Mellor DJ et al (2020) The 2020 Five Domains Model: Including Human-Animal Interactions in Assessments of Animal Welfare. Animals 10, 1870 https://www.ncbi.nlm.nih.gov/pmc/articles/PMC7602120/

Line 106 Typo ‘glucocorticoids’ instead of ‘glucokortycoids’

Line 119 Known population density? Climate zone? Terrain? Landscape & predominant vegetation type(s)? Many readers will also be unfamiliar with the principles of these specific hunts so can you provide more detail about why these harvests were undertaken (e.g., recreational, population management), how long the hunting season goes for, were the hunters employees of the Forest District/licensed hunters/any member of the public? It would also be advisable to include more detail about the specific methods used by the hunters who killed the deer in this study (e.g., on foot/ in a plane/vehicle, one person/small groups/large hunting parties, +/- dogs, typically how long deer were stalked, calibre of weapons & ammunition specs etc).

Line 125 I am incredibly concerned by this statement - “Ethical review and approval does not apply for this study”. It is no reflection on the individual authors if this is the case under current legislation in Poland but this is not consistent with ethical use of animals in research in other jurisdictions.

A study which includes animals being hunted and shot, whether that be solely for study purposes, as part of routine population management activities or recreational shooting, should require institutional Animal Ethics Committee approval.

For example, this study that piggy backed on aerial shooting trials being conducted on a conservation estate in Australia had to have approval by the relevant institutional Animal Ethics Committee. Hampton et al (2021) A comparison of fragmenting lead-based and lead-free bullets for aerial shooting of wild pigs. PLOS One https://journals.plos.org/plosone/article?id=10.1371/journal.pone.0247785

I note MDPI policy that – “If ethical approval is not required by national laws, authors must provide an exemption from the ethics committee, if one is available. Where a study has been granted exemption, the name of the ethics committee that provided this should be stated in Section ‘Institutional Review Board Statement’ with a full explanation on why the ethical approval was not required. If no animal ethics committee is available to review applications, authors should be aware that the ethics of their research will be evaluated by reviewers and editors”

I would strongly urge the Editor and the authors to ensure that any future studies of this nature go through an Animal Ethics Committee approval process, regardless of whether that is required by law or not. The excuse that “these activities are happening anyway” is just not good enough. Yes, there are many things being done to animals in the world but they are not being published in a leading international journal. To uphold standards of scientific ethics and integrity, it is critical that any peer-reviewed publication that involves the use of animals goes through AEC review.

Line 17 Can you specify how it was known that Polish Hunting Law was complied with? E.g., were inspectors observing the hunt?

Line 130 What is meant by ‘immediately’? e.g., Within 3 minutes? Within 5 minutes?
Are you able to give approximate times of day (diurnal fluctuations as you’ve mentioned in your discussion)?
Are you able to provide any description of the behavioural responses of deer (e.g., did their head/ear/tail carriage change? did they panic & flee, did they freeze)
Were full post-mortems conducted? In which case, are you able to comment on body condition, coat condition, injuries, repro status, any indications of gross pathology etc?

Line 209 It is not only ‘increased physical activity’ that explains increased CORT in this context but also pain, fear, panic, distress etc. It would be pertinent here to acknowledge not only the physical and physiological stress but also the psychological stress manifest in your CORT results. I’m sure you’re also familiar with the extensive literature on stress in red deer and other hunted species and it would strengthen your discussion to incorporate more of this here e.g., Viela et al (2020) Physiological stress reactions in red deer induced by hunting activities. Animals 10, 6, 1003 https://www.ncbi.nlm.nih.gov/pmc/articles/PMC7341308/

Line 211 Is this comparing apples with apples? Did the study on car collision use the same methodology (blood collection method, ELISA kit etc.) that you did?

Line 216 Are you able to draw any further meaning from the lack of significant differences in CORT between age & sex cohorts? E.g., Do you think this could indicate that regardless of age or sex, being hunted is a stressful experience for red deer.

Line 219 Discussion of diurnal and seasonal variations are always relevant but do you think that the acute stress of hunting would overshadow any of those more variations?

Line 226 Besides mating season, do you think there could be any other alternative explanations for your mean CORT results being higher than these other studies you mention? Given they focused on unrestrained farmed deer, compared to your study which focused on deer subjected to severe acute stress?

Line 228 Are you able to compare your blood biochem results to any known reference ranges and comment on the overall health status of the deer in the sample/in this population?

Line 261 It is concerning to hear that “some individuals probably did not die immediately after being shot” as this represents a significant animal welfare issue, which would violate mandated codes of practice in other jurisdictions. This does not meet the definition for humane killing https://kb.rspca.org.au/knowledge-base/what-does-the-term-humane-killing-or-humane-slaughter-mean/

Can you provide comment on whether particularly high CORT was noted in individuals who were not shot outright?

Line 282 Can you clarify here what you mean by “the present study showed a decrease in the UREA levels”? Decreased compared to what? Did you have baseline values to compare to?

Line 295 “over 80% of females could have been pregnant” This is another significant animal welfare concern and would not be permitted in other jurisdictions. For example under Wildlife (General) Regulations 2010 in Tasmania, Australia licenses to shoot deer for crop protection are generally not issued when females are pregnant or have dependent young https://pestsmart.org.au/wp-content/uploads/sites/3/2023/06/National-Code-of-Practice-for-Feral-and-Wild-Deer.pdf and as per the Code of Practice for the Hunting of Wild Fallow Deer in Tasmania (2012)
“Female deer are not generally permitted to be taken during the period November to mid-March to avoid adverse welfare outcomes related to shooting heavily pregnant does and does with dependent young” https://ablis.business.gov.au/service/tas/code-of-practice-for-the-hunting-of-wild-fallow-deer-in-tasmania/31907

Line 343 You’ve concluded with “improving hunting management” but you’ve not included any details about the management and methods in your introduction, methods or discussion and in your interpretation of your results, you seem to focus mainly on the influence of the mating season & nutrition rather than the influence of hunting management & methods. In your introduction and/or methods sections you need to add much more detail about how the hunt was conducted and, in your discussion, incorporate further thoughts on the influence of hunting management & methods. Otherwise, as it stands, you leave the reader guessing as to how your results relate to the specific hunting management & methods used.

Overall comments

- I completely understand that the authors are not responsible for the way the hunt was conducted. However, the hunting methods used constitute study methodology that must be accounted for if this paper is to be published. It is essential that issues raised in the paper, including failure to kill deer outright and killing pregnant females and their unborn fawns, are highlighted as significant animal welfare concerns that entail unacceptable animal suffering and would be contrary to legislation in other jurisdictions.  

- I believe you found mean age 6yrs (male), 4yrs (female) – if this is typical of the age cohort of deer being shot in this population are you able to provide any comment on how this might be affecting the demographics of that population? And if so, whether social disruption may be another potential stressor? For example, in other highly social species (albeit in an entirely different taxa) social stability is threatened by hunting pressure (e.g., Ben-Ami et al (2014) The welfare ethics of the commercial killing of free-ranging kangaroos: an evaluation of the benefits and costs of the industry. Animal Welfare 23, 1-10 https://www.cambridge.org/core/journals/animal-welfare/article/abs/welfare-ethics-of-the-commercial-killing-of-freeranging-kangaroos-an-evaluation-of-the-benefits-and-costs-of-the-industry/276694F7D915166A7CDB9CCA06806F82

Author Response

Reviewer 3

In many places in the world, different species of deer are hunted for various reasons. As such this paper will be of interest to a wide audience. In addition, this is an important topic to confront because deer hunting exposes animals to pain, fear, stress, distress and, by definition, premature death. As such, this study makes an important contribution to the literature because it demonstrates the stress caused by stalking hunts.

Response: Thank you for your kind comments.

However, the manuscript raises several serious animal welfare concerns including failure to kill animals outright and the killing of pregnant females and unborn fawns, which would be contrary to legislation in other jurisdictions. It is my strong view that while the manuscript has merit, these and several other aspects of the paper require major revision to be considered for publication.

Response:  Due to the fact that the red deer (Cervus elaphus) were shooted in accordance with the Polish Hunting Law (Annex to Resolution No. 57/2005 of February 22, 2005) during the hunting season (from September 1 to December 31, 2022) while stalking hunts (without the use of dogs, cars,  without drive animals), by qualified hunters - employees of the Lubartów forest district and no research procedures detrimental to animal welfare were performed, according to the applicable legislation (Act of 15 January 2015 on the protection of animals used for scientific or educational purposes; Directive 2010/63/EU of the European Parliament and of the Council of 22 September 2010 on the protection of animals used for scientific purposes), the consent of the ethics committee for the study was not required (Animal Welfare Commision Faculty of Animal Sciences and Bioeconomy at the Univerity of Life Sciences in Lublin, ZdsDz/6/2023).

Introduction
Line 33 This is interesting background information particularly to readers who are unfamiliar with deer hunting in Poland and Europe more broadly. Suggest you expand to include several paragraphs:
- General intro on numbers & species (as you have provided lines 33-35)

Response: It was added, please see lines 48-52.

- Hunting pressures, trends, methods, controversies in Poland & elsewhere
- The suite of potential stressors associated with hunting (line 109 you mention short-term stress at the time of shooting and long-term stress caused by presence of hunters but there are so many more potential stressors to both the hunted and the survivors (e.g., wounding, social disruption interrupted feeding and drinking etc.). If your central proposition is that [CORT] in blood collected immediately post-kill is an accurate reflection of the severity of acute stress, it’s highly relevant for readers to understand all the potential stressors that could be contributing to the CORT spike you predict.

Response: It was added, please see lines 59-71.

- The sensitivity of deer to negative stimuli including those associated with hunting: this is a key point that deserves a paragraph in your introduction. Draw on literature about behavioural reactivity of cervids e.g., freeze, flight, their propensity to panic. Brief statement of the relevant behavioural ecology would be valuable too i.e., why deer are like this. Anyone who has worked with captive or wild deer has seen them react so dramatically to stimuli that they’ve caused themselves fatal injuries like running into walls, fences etc. 
- Stress physiology (as you have provided lines 39 onwards)

Response: It was added, please see lines: 59-71

Line 49 Typo ‘in the field’ instead of ‘in the filed’

Response: It was improved, please see line 80.

Line 61 Would be pertinent to add ‘exercise’ or ‘physical exertion’ to this list as you go on to discuss this later in the paper 

Response: It was improved, please see line 109.

Line 87 It is meaningful here that you have highlighted CORT as an animal welfare indicator but this section would be strengthened if you not only talk about physical state e.g., (“potential disease states”) but also the impact on the animal’s behavioural and mental state i.e., the Five Domains of Animal Welfare 
Mellor DJ et al (2020) The 2020 Five Domains Model: Including Human-Animal Interactions in Assessments of Animal Welfare. Animals 10, 1870 https://www.ncbi.nlm.nih.gov/pmc/articles/PMC7602120/
Response: It was added, please see lines 122-126.

Line 106 Typo ‘glucocorticoids’ instead of ‘glucokortycoids’

Response: It was improved, please see line 143.

Line 119 Known population density? Climate zone? Terrain? Landscape & predominant vegetation type(s)? Many readers will also be unfamiliar with the principles of these specific hunts so can you provide more detail about why these harvests were undertaken (e.g., recreational, population management), how long the hunting season goes for, were the hunters employees of the Forest District/licensed hunters/any member of the public? It would also be advisable to include more detail about the specific methods used by the hunters who killed the deer in this study (e.g., on foot/ in a plane/vehicle, one person/small groups/large hunting parties, +/- dogs, typically how long deer were stalked, calibre of weapons & ammunition specs etc).

Response: It was added, please see lines 156-178.

Line 125 I am incredibly concerned by this statement - “Ethical review and approval does not apply for this study”. It is no reflection on the individual authors if this is the case under current legislation in Poland but this is not consistent with ethical use of animals in research in other jurisdictions.

A study which includes animals being hunted and shot, whether that be solely for study purposes, as part of routine population management activities or recreational shooting, should require institutional Animal Ethics Committee approval.

For example, this study that piggy backed on aerial shooting trials being conducted on a conservation estate in Australia had to have approval by the relevant institutional Animal Ethics Committee. Hampton et al (2021) A comparison of fragmenting lead-based and lead-free bullets for aerial shooting of wild pigs. PLOS One https://journals.plos.org/plosone/article?id=10.1371/journal.pone.0247785

I note MDPI policy that – “If ethical approval is not required by national laws, authors must provide an exemption from the ethics committee, if one is available. Where a study has been granted exemption, the name of the ethics committee that provided this should be stated in Section ‘Institutional Review Board Statement’ with a full explanation on why the ethical approval was not required. If no animal ethics committee is available to review applications, authors should be aware that the ethics of their research will be evaluated by reviewers and editors”

I would strongly urge the Editor and the authors to ensure that any future studies of this nature go through an Animal Ethics Committee approval process, regardless of whether that is required by law or not. The excuse that “these activities are happening anyway” is just not good enough. Yes, there are many things being done to animals in the world but they are not being published in a leading international journal. To uphold standards of scientific ethics and integrity, it is critical that any peer-reviewed publication that involves the use of animals goes through AEC review.

Response: Due to the fact that the red deer (Cervus elaphus) were shooted in accordance with the Polish Hunting Law (Annex to Resolution No. 57/2005 of February 22, 2005) during the hunting season (from September 1 to December 31, 2022) while stalking hunts (without the use of dogs, cars,  without drive animals), by qualified hunters - employees of the Lubartów forest district and no research procedures detrimental to animal welfare were performed, according to the applicable legislation (Act of 15 January 2015 on the protection of animals used for scientific or educational purposes; Directive 2010/63/EU of the European Parliament and of the Council of 22 September 2010 on the protection of animals used for scientific purposes), the consent of the ethics committee for the study was not required (Animal Welfare Commission Faculty of Animal Sciences and Bioeconomy at the University of Life Sciences in Lublin, ZdsDz/6/2023). We have attached the required document to the research.

Line 17 Can you specify how it was known that Polish Hunting Law was complied with? E.g., were inspectors observing the hunt?

Response: It was added, please see lines 156-168.

Line 130 What is meant by ‘immediately’? e.g., Within 3 minutes? Within 5 minutes?
Are you able to give approximate times of day (diurnal fluctuations as you’ve mentioned in your discussion)?

Response: It was added, please see line 199.

Are you able to provide any description of the behavioural responses of deer (e.g., did their head/ear/tail carriage change? did they panic & flee, did they freeze)
Were full post-mortems conducted? In which case, are you able to comment on body condition, coat condition, injuries, repro status, any indications of gross pathology etc?

Response: It was added, please see lines 156-168.

Line 209 It is not only ‘increased physical activity’ that explains increased CORT in this context but also pain, fear, panic, distress etc. It would be pertinent here to acknowledge not only the physical and physiological stress but also the psychological stress manifest in your CORT results. I’m sure you’re also familiar with the extensive literature on stress in red deer and other hunted species and it would strengthen your discussion to incorporate more of this here e.g., Viela et al (2020) Physiological stress reactions in red deer induced by hunting activities. Animals 10, 6, 1003 https://www.ncbi.nlm.nih.gov/pmc/articles/PMC7341308/

Response: It was added, please see lines 282-287.

Line 211 Is this comparing apples with apples? Did the study on car collision use the same methodology (blood collection method, ELISA kit etc.) that you did?

Response: In this sentences we wanted to demonstrate the power of other negative stress stimuli.

Line 216 Are you able to draw any further meaning from the lack of significant differences in CORT between age & sex cohorts? E.g., Do you think this could indicate that regardless of age or sex, being hunted is a stressful experience for red deer.

Response: Yes we agree with the sentence: "could indicate that regardless of age or sex, being hunted is a stressful experience for red deer", and it was added, please see lines 290-291.

Line 219 Discussion of diurnal and seasonal variations are always relevant but do you think that the acute stress of hunting would overshadow any of those more variations?
Response: Most likely yes because of the loss of life, which is the most stressful situation.

Line 226 Besides mating season, do you think there could be any other alternative explanations for your mean CORT results being higher than these other studies you mention? Given they focused on unrestrained farmed deer, compared to your study which focused on deer subjected to severe acute stress?
Response: The mating season in deer is very strongly demonstrated by animals and is associated with very large behavioural and physiological changes, so in our opinion it is rather the main cause.

Line 228 Are you able to compare your blood biochem results to any known reference ranges and comment on the overall health status of the deer in the sample/in this population?

Response: Yes, it was added, please see table 6.

Line 261 It is concerning to hear that “some individuals probably did not die immediately after being shot” as this represents a significant animal welfare issue, which would violate mandated codes of practice in other jurisdictions. This does not meet the definition for humane killing https://kb.rspca.org.au/knowledge-base/what-does-the-term-humane-killing-or-humane-slaughter-mean/

Response: It has been deleted. For the study, we selected animals that died immediately after being shot.

Can you provide comment on whether particularly high CORT was noted in individuals who were not shot outright?

Response: This may most likely be related to the time taken to collect blood for testing, i.e. up to 5 minutes after the animal's death. This could have influenced the variability of the obtained results.

Line 282 Can you clarify here what you mean by “the present study showed a decrease in the UREA levels”? Decreased compared to what? Did you have baseline values to compare to?
Response: It was improved, “In the present study it showed much higher values in the UREA of all the analysed red deer compared to the results obtained by Rosef et al. [87] u free-ranging animals, which may be related to the insufficient protein intake due to the reduction of appetite during the mating season [92].” Please see lines 360-364.

Line 295 “over 80% of females could have been pregnant” This is another significant animal welfare concern and would not be permitted in other jurisdictions. For example under Wildlife (General) Regulations 2010 in Tasmania, Australia licenses to shoot deer for crop protection are generally not issued when females are pregnant or have dependent young https://pestsmart.org.au/wp-content/uploads/sites/3/2023/06/National-Code-of-Practice-for-Feral-and-Wild-Deer.pdf and as per the Code of Practice for the Hunting of Wild Fallow Deer in Tasmania (2012)
“Female deer are not generally permitted to be taken during the period November to mid-March to avoid adverse welfare outcomes related to shooting heavily pregnant does and does with dependent young” https://ablis.business.gov.au/service/tas/code-of-practice-for-the-hunting-of-wild-fallow-deer-in-tasmania/31907
Response: The hinds were hunted from September 1 to October 31, just after the mating season, which means they could become pregnant, but it was a very early pregnancy.

Line 343 You’ve concluded with “improving hunting management” but you’ve not included any details about the management and methods in your introduction, methods or discussion and in your interpretation of your results, you seem to focus mainly on the influence of the mating season & nutrition rather than the influence of hunting management & methods. In your introduction and/or methods sections you need to add much more detail about how the hunt was conducted and, in your discussion, incorporate further thoughts on the influence of hunting management & methods. Otherwise, as it stands, you leave the reader guessing as to how your results relate to the specific hunting management & methods used.

Response: It was added, please see lines 156-192, 412-428.

Overall comments

- I completely understand that the authors are not responsible for the way the hunt was conducted. However, the hunting methods used constitute study methodology that must be accounted for if this paper is to be published. It is essential that issues raised in the paper, including failure to kill deer outright and killing pregnant females and their unborn fawns, are highlighted as significant animal welfare concerns that entail unacceptable animal suffering and would be contrary to legislation in other jurisdictions.

Response: Due to the fact that the red deer (Cervus elaphus) were shooted in accordance with the Polish Hunting Law (Annex to Resolution No. 57/2005 of February 22, 2005) during the hunting season while stalking hunts (without the use of dogs, cars,  without drive animals), by qualified hunters - employees of the Lubartów forest district and no research procedures detrimental to animal welfare were performed, according to the applicable legislation (Act of 15 January 2015 on the protection of animals used for scientific or educational purposes; Directive 2010/63/EU of the European Parliament and of the Council of 22 September 2010 on the protection of animals used for scientific purposes), the consent of the ethics committee for the study was not required (Animal Welfare Commision Faculty of Animal Sciences and Bioeconomy at the Univerity of Life Sciences in Lublin, ZdsDz/6/2023).

- I believe you found mean age 6yrs (male), 4yrs (female) – if this is typical of the age cohort of deer being shot in this population are you able to provide any comment on how this might be affecting the demographics of that population? And if so, whether social disruption may be another potential stressor? For example, in other highly social species (albeit in an entirely different taxa) social stability is threatened by hunting pressure (e.g., Ben-Ami et al (2014) The welfare ethics of the commercial killing of free-ranging kangaroos: an evaluation of the benefits and costs of the industry. Animal Welfare 23, 1-10 https://www.cambridge.org/core/journals/animal-welfare/article/abs/welfare-ethics-of-the-commercial-killing-of-freeranging-kangaroos-an-evaluation-of-the-benefits-and-costs-of-the-industry/276694F7D915166A7CDB9CCA06806F82

Response: We didn't have the opportunity to choose which animals we caught from hunting, and such a comment may have been too far-fetched. We agree that hunting may change the age structure of animals herd, because, for example, game management is allowed, which is associated with the aging or rejuvenation of the population. However, this study did not address this. That's why we didn't add this information in the introduction. We will certainly take this into account in the future and expand our research with such data.

Round 2

Reviewer 2 Report

Comments and Suggestions for Authors

Your responses did not really answer my comments but still the manuscript improved as somehow you included a table comparing previous studies (including populations without stalking hunting pressure) and that is more or less what I asked to do. In the original version you could not answer your aim as you had no contrasting information, no comparisons. Now you added the table in the discussion and the manuscript improved. You should have integrate that table better, for example adding the data collected in your study as well in there for a quick comparison by the reader. 

Regarding the comment on the stats, you are saying GLMs are not used since factors are continuous. You need to be aware that GLMs allows for both fixed and continuous factors. Here there are two options, use GLMs and put all factors in one model, or correct the p-value for multiple hypotheses testing on the same dependent variable. Otherwise you increase the type I error rate. 

The introduction is too focused on deer in Poland, this limits the chances of researchers working in other contexts to cite this paper. It would be much more interesting to have hunting/stalking and relative stress as broader topic before going to the specific context.  

Author Response

Reviewer 2

Your responses did not really answer my comments but still the manuscript improved as somehow you included a table comparing previous studies (including populations without stalking hunting pressure) and that is more or less what I asked to do. In the original version you could not answer your aim as you had no contrasting information, no comparisons. Now you added the table in the discussion and the manuscript improved. You should have integrate that table better, for example adding the data collected in your study as well in there for a quick comparison by the reader.

Response: We sincerely would like to thank the Reviewer for the time and commitment spent on our work. As suggested, we have made corrections to improve the quality of our manuscript. All corrections are marked by text tracking. Table 6 has been supplemented with data collected in our study. Please see Table 6.

Regarding the comment on the stats, you are saying GLMs are not used since factors are continuous. You need to be aware that GLMs allows for both fixed and continuous factors. Here there are two options, use GLMs and put all factors in one model, or correct the p-value for multiple hypotheses testing on the same dependent variable. Otherwise you increase the type I error rate.

Response: The dependencies in Tables 2-5 were analyzed due to the distribution of variables and were analyzed using non-parametric tests; consequently, it is not possible to use GLM for these variables - they deviate from the normal distribution. Referring to Table 1, we analyzed the effect of sex (fixed factor) and the effect of carcass weight and age (regression factors) in 1 GLM model on plasma biochemical parameters. The analysis only confirmed the effect of gender on HDL cholesterol, while the regression of age and carcass weight on all characteristics had no significant effect. Moreover, only four features out of 10 can be analyzed with a parametric model. Thank you for your suggestion, we apologize that we did not use the GLM model for calculations, we will try to change this in future research. The GLM analysis is attached below.

The introduction is too focused on deer in Poland, this limits the chances of researchers working in other contexts to cite this paper. It would be much more interesting to have hunting/stalking and relative stress as broader topic before going to the specific context.

Response: It has been added, please see lines: 48-54.

Reviewer 3 Report

Comments and Suggestions for Authors

The authors have made a concerted effort to address my comments. I understand they will have been given a limited amount of time to respond. As such, there are a few typos that need to be addressed, and one or two minor areas which I feel still require some further work.

In regards to animal welfare and ethics concerns, I appreciate your efforts to explain Polish legislation and the provision of a letter from the Chair of the Animal Welfare Commission. While I do not hold the authors responsible, I still maintain that there are serious animal welfare concerns with how the hunt was conducted - some animals not killed outright & as many as 80% of females pregnant - which would be contrary to legislation in other jurisdictions. However, weighing up the detriment vs the benefit of publishing vs rejecting this paper, I feel the greater animal welfare benefit will be from its publication. 

Minor corrections incl:

Line 59 Typo “The welfare of deer animals” – choose either ‘deer’ or ‘animals’

Line 59 to 70 This additional section is missing critical information about how human activities, in the context of this study namely hunting methods, cause stress to deer.

Line 91 ‘Severity’ instead of ‘stringency’?

Line 124 In my original comments, I suggested the authors include some introduction to the Five Domains. Equating blood biochemical parameters to the Five Domains is inaccurate. Physiological health is distinct from animal welfare. I’d urge the authors to take a little more time to addresses the original comment as I feel it would strengthen the manuscript to show how their findings relate to the Five Domains of Animal Welfare.

Line 130 ‘showed no testimony’ isn’t quite the right phrase. Maybe ‘found no evidence of’?

Line 135 Typo ‘lasnt’?

Line 152 to 164 Thank you to the authors for including further information about the hunting methods used as this is highly relevant information for the reader.

Line 161 Typo “the hunt” instead of “huntings”

Line 265-270, 272-274 Important & interesting additions

Line 295 Typo “Two different” instead of “two difrent”

Line 325 Typo “Anaesthetic” instead of “aneesthetic”

Line 341 Typo “u free-ranging animals” delete “u”?

Line 343 Typo “[92]. s.”?

Comments on the Quality of English Language

Just a few typos. 

Author Response

Reviewer 3

The authors have made a concerted effort to address my comments. I understand they will have been given a limited amount of time to respond. As such, there are a few typos that need to be addressed, and one or two minor areas which I feel still require some further work.

In regards to animal welfare and ethics concerns, I appreciate your efforts to explain Polish legislation and the provision of a letter from the Chair of the Animal Welfare Commission. While I do not hold the authors responsible, I still maintain that there are serious animal welfare concerns with how the hunt was conducted - some animals not killed outright & as many as 80% of females pregnant - which would be contrary to legislation in other jurisdictions. However, weighing up the detriment vs the benefit of publishing vs rejecting this paper, I feel the greater animal welfare benefit will be from its publication. 
Response: We sincerely would like to thank the Reviewer for the time and commitment spent on our work. As suggested, we have made corrections to improve the quality of our manuscript. All corrections are marked by text tracking.

Minor corrections incl:

Line 59 Typo “The welfare of deer animals” – choose either ‘deer’ or ‘animals’

Line 59 to 70 This additional section is missing critical information about how human activities, in the context of this study namely hunting methods, cause stress to deer.

Line 91 ‘Severity’ instead of ‘stringency’?

Line 124 In my original comments, I suggested the authors include some introduction to the Five Domains. Equating blood biochemical parameters to the Five Domains is inaccurate. Physiological health is distinct from animal welfare. I’d urge the authors to take a little more time to addresses the original comment as I feel it would strengthen the manuscript to show how their findings relate to the Five Domains of Animal Welfare.

Response: It has been improved, please see lines: 125-134.

Line 130 ‘showed no testimony’ isn’t quite the right phrase. Maybe ‘found no evidence of’?

Line 135 Typo ‘lasnt’?

Line 152 to 164 Thank you to the authors for including further information about the hunting methods used as this is highly relevant information for the reader.

Line 161 Typo “the hunt” instead of “huntings”

Line 265-270, 272-274 Important & interesting additions

Line 295 Typo “Two different” instead of “two difrent”

Line 325 Typo “Anaesthetic” instead of “aneesthetic”

Line 341 Typo “u free-ranging animals” delete “u”?

Line 343 Typo “[92]. s.”?
